The EMBO Journal (2013) 32, 2321–2335
www.embojournal.org

# Sir2 is required for Clr4 to initiate centromeric heterochromatin assembly in fission yeast

Benjamin J Alper[1], Godwin Job[1], Rajesh K Yadav[1], Sreenath Shanker[1], Brandon R Lowe[1,2] and Janet F Partridge[1,2,*]

[1]Department of Biochemistry, St Jude Children's Research Hospital, Memphis, TN, USA and [2]Integrated Program in Biomedical Sciences, University of Tennessee Health Science Center, Memphis, TN, USA

Heterochromatin assembly in fission yeast depends on the Clr4 histone methyltransferase, which targets H3K9. We show that the histone deacetylase Sir2 is required for Clr4 activity at telomeres, but acts redundantly with Clr3 histone deacetylase to maintain centromeric heterochromatin. However, Sir2 is critical for Clr4 function during *de novo* centromeric heterochromatin assembly. We identified new targets of Sir2 and tested if their deacetylation is necessary for Clr4-mediated heterochromatin establishment. Sir2 preferentially deacetylates H4K16Ac and H3K4Ac, but mutation of these residues to mimic acetylation did not prevent Clr4-mediated heterochromatin establishment. Sir2 also deacetylates H3K9Ac and H3K14Ac. Strains bearing H3K9 or H3K14 mutations exhibit heterochromatin defects. H3K9 mutation blocks Clr4 function, but why H3K14 mutation impacts heterochromatin was not known. Here, we demonstrate that recruitment of Clr4 to centromeres is blocked by mutation of H3K14. We suggest that Sir2 deacetylates H3K14 to target Clr4 to centromeres. Further, we demonstrate that Sir2 is critical for *de novo* accumulation of H3K9me2 in RNAi-deficient cells. These analyses place Sir2 and H3K14 deacetylation upstream of Clr4 recruitment during heterochromatin assembly.

*The EMBO Journal* (2013) **32**, 2321–2335. doi:10.1038/emboj.2013.143; Published online 14 June 2013
Subject Categories: chromatin & transcription
Keywords: centromere; fission yeast; heterochromatin; histone deacetylase; histone methyltransferase

## Introduction

In fission yeast, repressive chromatin assembles at the centromeres, mating type locus and at telomeres (Klar and Bonaduce, 1991; Thon and Klar, 1992; Allshire *et al*, 1994; Nimmo *et al*, 1994; Allshire *et al*, 1995). This heterochromatin is critical for the maintenance of genomic integrity and to determine cell type identity, and is characterized by both hypoacetylation of the histone tails

*Corresponding author. Department of Biochemistry, St Jude Children's Research Hospital, 262 Danny Thomas Place, Memphis, TN 38105, USA. Tel.: +1 901 595 2679; Fax: +1 901 525 8025; E-mail: janet.partridge@stjude.org

(Ekwall *et al*, 1997; Bjerling *et al*, 2002; Gomez *et al*, 2005) and by methylation of histone H3 on lysine 9 (Bannister *et al*, 2001; Nakayama *et al*, 2001; Cam *et al*, 2005). Several deacetylases contribute to histone hypoacetylation, including Clr3 and Clr6 (Bjerling *et al*, 2002), as well as members of a distinct class of deacetylases; the Sir2 family (Shankaranarayana *et al*, 2003; Freeman-Cook *et al*, 2005).

The Sir2 family is a group of $NAD^+$-dependent lysine deacetylases that are conserved among eukaryotes and archaea (Brachmann *et al*, 1995; Imai *et al*, 2000; Landry *et al*, 2000; Smith *et al*, 2000). Sir2 homologues are important for silencing of constitutive heterochromatin, and have both histone and non-histone substrates (Bell *et al*, 2002; Yang *et al*, 2005). In budding yeast, heterochromatin assembly at telomeres and at the mating type locus is reliant on Sir2p and other components of the SIR complex, and the role of these proteins in the assembly of silent chromatin has been well studied (Johnson *et al*, 1990; Aparicio *et al*, 1991; Braunstein *et al*, 1993; Johnson *et al*, 2009; Hickman *et al*, 2011; Oppikofer *et al*, 2011). In the simplest model, silencer proteins bind to specific DNA sequences located within the telomere repeat sequences or at silencers within the mating type region and directly recruit the SIR silencing complex to these initiation sites. Next, histone deacetylation by Sir2 facilitates binding of Sir3 and Sir4 to proximal nucleosomes, which in turn allows spreading of the repressive complex (reviewed in Rusche *et al* (2003)).

In contrast to budding yeast, how fission yeast homologues of Sir2 promote heterochromatin assembly is comparatively understudied. For example, it is not known whether Sir2 proteins in fission yeast are components of structural silencing complexes like the SIR complex, or whether these proteins are recruited to genomic regions by specific binding factors to promote heterochromatin initiation. The Sir2 family in fission yeast comprises three members, namely Sir2, Hst2 and Hst4. Fission yeast Sir2 is a primarily nuclear protein that is important for the silencing of telomeres and the mating type locus, and also plays a role in centromeric silencing (Shankaranarayana *et al*, 2003; Freeman-Cook *et al*, 2005), although the extent of its influence at centromeres is less clear. The fission yeast Sir2 homologue, Hst4, also has roles in heterochromatic silencing (Freeman-Cook *et al*, 1999). Hst4 has deacetylase activity on H3K56Ac (Haldar and Kamakaka, 2008), and is involved in repair of DNA damage during S phase, while Hst2 is generally less well characterized.

Confusion persists about the role of fission yeast Sir2 in centromeric silencing. While it is clear that Sir2 plays an important role at telomeres and at the mating type locus, the silencing functions of Sir2 at centromeres are less pronounced and are dependent on the location of the reporter gene (Shankaranarayana *et al*, 2003; Freeman-Cook *et al*, 2005). H3K9Ac peptides have been shown to serve as Sir2 targets *in vitro* (Shankaranarayana *et al*, 2003). Consistent with this observation, H3K9Ac is increased at

heterochromatic loci in *sir2* mutants, which may or may not lead to a reduction in H3K9me2 and a reduction in Swi6 localization to centromeres (Shankaranarayana *et al*, 2003; Freeman-Cook *et al*, 2005).

In this study, we have evaluated centromeric functions of Sir2 in greater detail. We show that Sir2 functions upstream of and is required to promote *de novo* Clr4-mediated H3K9 methyltransferase activity, and that disruption of this pathway has physiological consequences consistent with the loss of centromere function. Further, we have probed the specific catalytic role of the Sir2 histone deacetylase activity in promoting Clr4 activity. We created a novel mutant of fission yeast Sir2 (sir2N247A) that is defective for deacetylase activity. We show that this mutant lacks deacetylase activity *in vitro* and *in vivo*, and largely fails to promote *de novo* function of Clr4 at centromeres. In addition, we examine the substrate specificity of Sir2 *in vitro*, and use this analysis to identify specific targets of Sir2 HDAC activity. We test whether these targets are important for establishing centromeric silencing.

## Results

### Sir2 HDAC mutants show a complete loss of Clr4 function at telomeres

To determine the contribution of Sir2's deacetylase activity to heterochromatin maintenance, we generated a mutation within the catalytic domain of Sir2 that changes a conserved asparagine to an alanine (N247A). The mutation resides in a conserved series of amino acids that make up the catalytic core of the enzyme, and is analogous to the N345A mutation in *Saccharomyces cerevisiae* Sir2p, which is a catalytically inactive protein both *in vivo* and *in vitro* (Imai *et al*, 2000; Armstrong *et al*, 2002; Oppikofer *et al*, 2011) (Figure 1A). The N247A mutation was engineered into the genomic *sir2* locus that we additionally marked with the *his3* allele and into strains that express a fully functional TAP-tagged allele of *sir2* at the endogenous locus. Western analyses showed that expression of the wild-type (WT) and N247A mutant Sir2–TAP proteins was equivalent (Figure 1B), suggestive that the N247A mutation does not alter Sir2 protein stability.

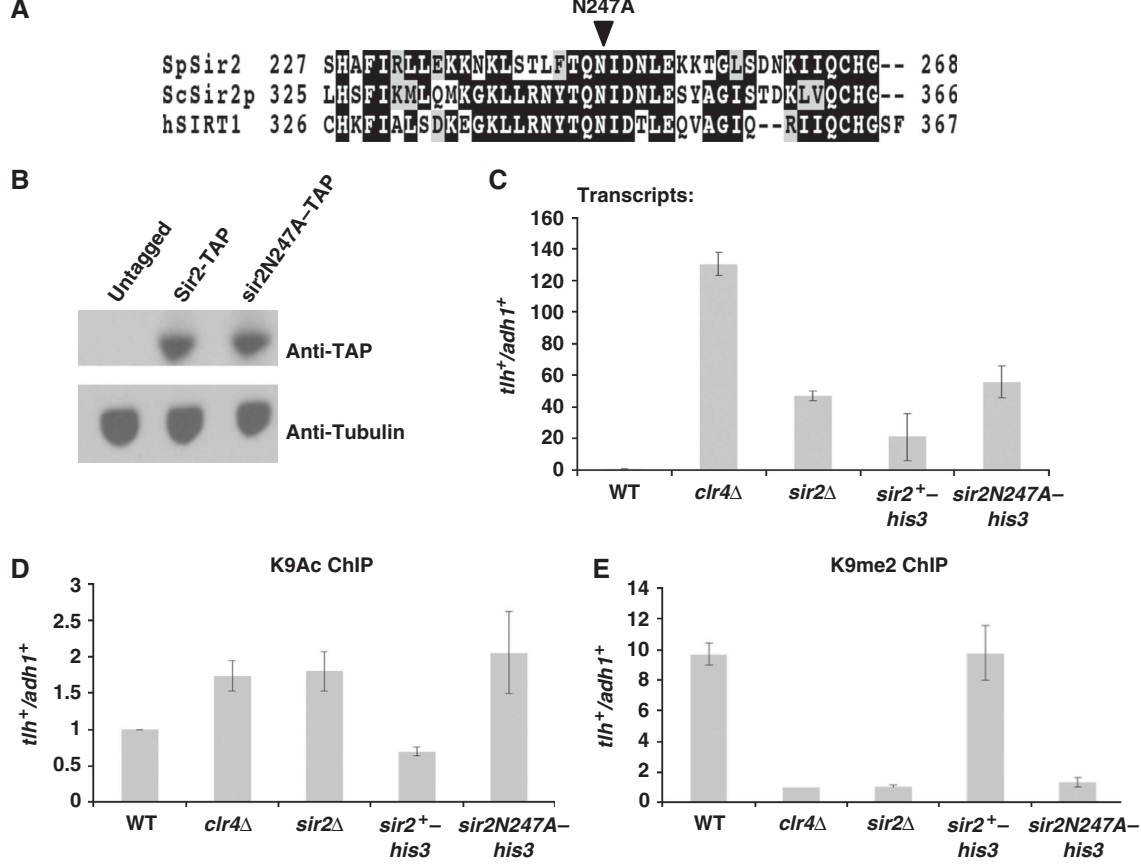

**Figure 1** The presumed catalytic mutant, sir2N247A, causes defects in the maintenance of subtelomeric heterochromatin. (**A**) Alignment of fission yeast Sir2 with nearest homologues from budding yeast and man. The fission yeast sir2N247A mutant is analogous to the budding yeast Sir2pN345A, which lacks catalytic activity. Position of the N247A substitution is indicated. (**B**) The predicted catalytic mutant Sir2N247A–TAP is stably expressed. Western blot of extracts from Sir2–TAP and sir2N247A–TAP expressing cells, probed for TAP with tubulin as loading control. (**C**) Subtelomeric transcripts accumulate in *sir2Δ* and *sir2N247A* mutant cells. mRNA transcripts from subtelomeric *tlh* genes were quantified by quantitative real-time PCR (qRT–PCR) amplification of cDNA, and normalized to transcript levels of the *adh1*$^+$ (alcohol dehydrogenase) control. Graphic data present the average of two distinct experimental replicates and error bars depict s.e.m. (**D**) H3K9 acetylation is increased and (**E**) H3K9me2 methylation is abolished at subtelomeres in *sir2* deletion and *sir2N247A* mutants. qRT–PCR analysis of ChIP experiments monitoring relative enrichment of *tlh* sequences over *adh1*$^+$ control euchromatic locus in immunoprecipitated samples. Graphs represent average of data obtained from two biological samples, with error bars depicting s.e.m.

We first analysed the effects of *sir2* deletion and the presumed catalytically dead *sir2N247A* mutation on gene silencing in subtelomeric regions (Figure 1C). In WT cells, subtelomeric heterochromatin coats and silences transcription from the *tlh* genes that are located close to the telomeres of the left arm of chromosome 1 and right arm of chromosome 2 (Hansen *et al*, 2006). Cells that lack *sir2* show accumulation of transcripts from *tlh* genes compared with WT cells, but this is not to the extent seen in *clr4Δ* cells, which exhibit very high levels of *tlh* transcripts. Importantly, the level of *tlh* transcript accumulation in the *sir2N247A* strain was similar to that in *sir2Δ* cells, suggesting that the *sir2N247A* mutant is defective for *sir2* function at telomeres.

We further tested the effects of mutation of Sir2 on subtelomeric heterochromatin maintenance. Sir2 has been reported to have H3K9 deacetylase activity in fission yeast (Shankaranarayana *et al*, 2003). Chromatin immunoprecipitation (ChIP) experiments performed with antibodies specific for H3K9Ac revealed that H3K9Ac levels are elevated on subtelomeric sequences in cells lacking *sir2*, and that the *sir2N247A* mutant causes the same enrichment of H3K9Ac as does loss of *sir2* (Figure 1D). We note that the level of enrichment for H3K9Ac on subtelomeric sequences is similar between *clr4Δ* and *sir2Δ* cells. Next, we examined H3K9me2 levels within the subtelomeric regions by ChIP (Figure 1E). Cells that lack *clr4* show a complete loss of H3K9me2 signal. We found that *sir2Δ* and the *sir2N247A* mutant both resemble *clr4Δ* cells, and appear to have complete loss of H3K9me2 from subtelomeric sequences. From these experiments, we conclude that the N247A mutant of Sir2 is impacting the catalytic activity of Sir2 *in vivo*. Critically, the loss of Sir2 or Sir2 HDAC activity results in a complete loss of H3K9me2 signal from the *tlh* genes, suggesting that Clr4 function is inhibited at subtelomeres.

### Loss of Sir2 HDAC activity does not affect maintenance of centromeric heterochromatin

Next, we examined the effect of *sir2Δ* and the *sir2N247A* mutant on centromeric heterochromatin. *sir2*-deficient fission yeast have been reported to exhibit different levels of defects in the maintenance of centromeric heterochromatin (Shankaranarayana *et al*, 2003; Freeman-Cook *et al*, 2005). We assessed the state of centromeric heterochromatin in strains lacking *sir2* or bearing the *sir2N247A* allele using a strain background that carries a *ura4⁺* transgene inserted within the outer repeats of the centromere (*cen::ura4⁺*) (Allshire *et al*, 1995). WT cells exhibit robust silencing of this centromeric transgene, allowing growth of cells on media that contains 5-fluoro-orotic acid (FOA), which is toxic to cells that express *ura4⁺* (Figure 2A). Cells that have defective centromeric heterochromatin, such as *clr4*-deficient cells, fail to grow on FOA. *sir2* null cells showed a mild defect in silencing of the transgene, such that while sufficient *ura4⁺* silencing occurs for cells to grow on FOA, there was enhanced growth on media lacking uracil, indicating some increase in the expression of the centromeric reporter. The *sir2N247A* mutant showed a minimal defect in silencing of the centromeric reporter similar to the *sir2* null.

Analysis of *cen::ura4⁺* transcripts and transcripts from the endogenous centromeric repeats (*dg* and *dh*) indicated that there was a very minor accumulation of transcripts in cells lacking *sir2*, especially when compared with *clr4Δ* cells, which accumulate high levels of centromeric transcripts (Figure 2B). Consistent with the maintenance of centromeric silencing, the RNAi pathway appears intact in *sir2Δ* cells since the processing of centromeric transcripts into siRNAs occurs efficiently (Figure 2C). In keeping with a lack of effect of *sir2* mutation on centromeric heterochromatin maintenance, we saw little evidence of *sir2Δ* cells displaying defects in chromosome segregation when cells were assessed for chromosome segregation defects during anaphase (Supplementary Figure S1). Loss of *sir2* does, however, impact the levels of H3K9 acetylation at centromeres. ChIP analysis revealed a three-fold increase in the levels of H3K9Ac on centromeric repeats in *sir2Δ* or the *sir2N247A* mutant cells, supporting a role for Sir2 in deacetylation of K9 of histone H3 at centromeres (Figure 2D) (Shankaranarayana *et al*, 2003).

Intriguingly, when we monitored H3K9me2 on centromeric repeats, we saw no loss of H3K9 methylation at centromeres in *sir2Δ* or *sir2N247A* cells (Figure 2E). This observation is in stark contrast to the effects at subtelomeres, where loss of *sir2* function caused a complete loss of H3K9me2. It suggests that although H3K9Ac is increased at centromeres, there is still sufficient H3K9 available for methylation by Clr4 at centromeres in *sir2*-deficient cells. In addition, while Sir2 and its HDAC activity are critical for Clr4 activity at subtelomeres, at centromeres, Sir2 is either not important for Clr4 function or acts redundantly with additional pathways to maintain centromeric heterochromatin.

We tested whether Sir2 acts redundantly in heterochromatin maintenance at centromeres with another histone deacetylase, Clr3 (Grewal *et al*, 1998). *clr3Δ* alone led to a minor defect in heterochromatin maintenance, but combining *clr3Δ* with either *sir2Δ* or *sir2N247A* led to a complete loss of centromeric silencing, and loss of centromeric heterochromatin as monitored by loss of recruitment of the Swi6 chromodomain protein to centromeres (Figure 2F–H). These data suggest that Sir2 and Clr3 play redundant roles in the maintenance of centromeric heterochromatin.

We have adopted a strategy to determine whether genes with minor roles in centromeric heterochromatin maintenance function upstream of Clr4 during heterochromatin initiation (Sadaie *et al*, 2004; Partridge *et al*, 2007). Removal of *clr4⁺* from WT cells causes complete loss of centromeric heterochromatin, with complete loss of H3K9me2, accumulation of high levels of centromeric transcripts, and high rates of chromosome missegregation. On reintegration of *clr4⁺* into its normal genomic location, there is a rapid and efficient establishment or initiation of heterochromatin, with resumption of silencing of centromeric transcripts and reassembly of H3K9me2 chromatin. However, if this *clr4* withdrawal and reintegration is performed in a genetic background lacking a factor required for the recruitment or *de novo* activity of Clr4 at centromeres, a defect in centromeric heterochromatin assembly is seen.

### Sir2 is required for heterochromatin reinitiation at centromeres

To test whether Sir2 plays a role in the initiation of centromeric heterochromatin, we asked whether *clr4⁺* could function when reintegrated back into its genomic locus in *sir2Δclr4Δ* cells (Figure 3A). As a control, *clr4⁺* was

reintegrated into cells lacking *clr4*$^+$ alone. Southern analysis was performed to confirm single-copy reintegration into the correct locus, and multiple independent reintegrant strains were analysed and data pooled.

In contrast to the efficient assembly of heterochromatin in *clr4Δ to clr4*$^+$ reintegrants, demonstrated by the resumption

of silencing of the centromeric transgene (Figure 3B), *sir2Δ clr4Δ* cells into which *clr4*$^+$ was reintegrated (*sir2Δ clr4Δ* to *sir2Δ clr4*$^+$ reintegrants) did not re-establish *cen::ura4*$^+$ transgene silencing, and grew on media lacking uracil, but not on media containing FOA. Analysis of centromeric transcripts confirmed that there was a failure to fully silence

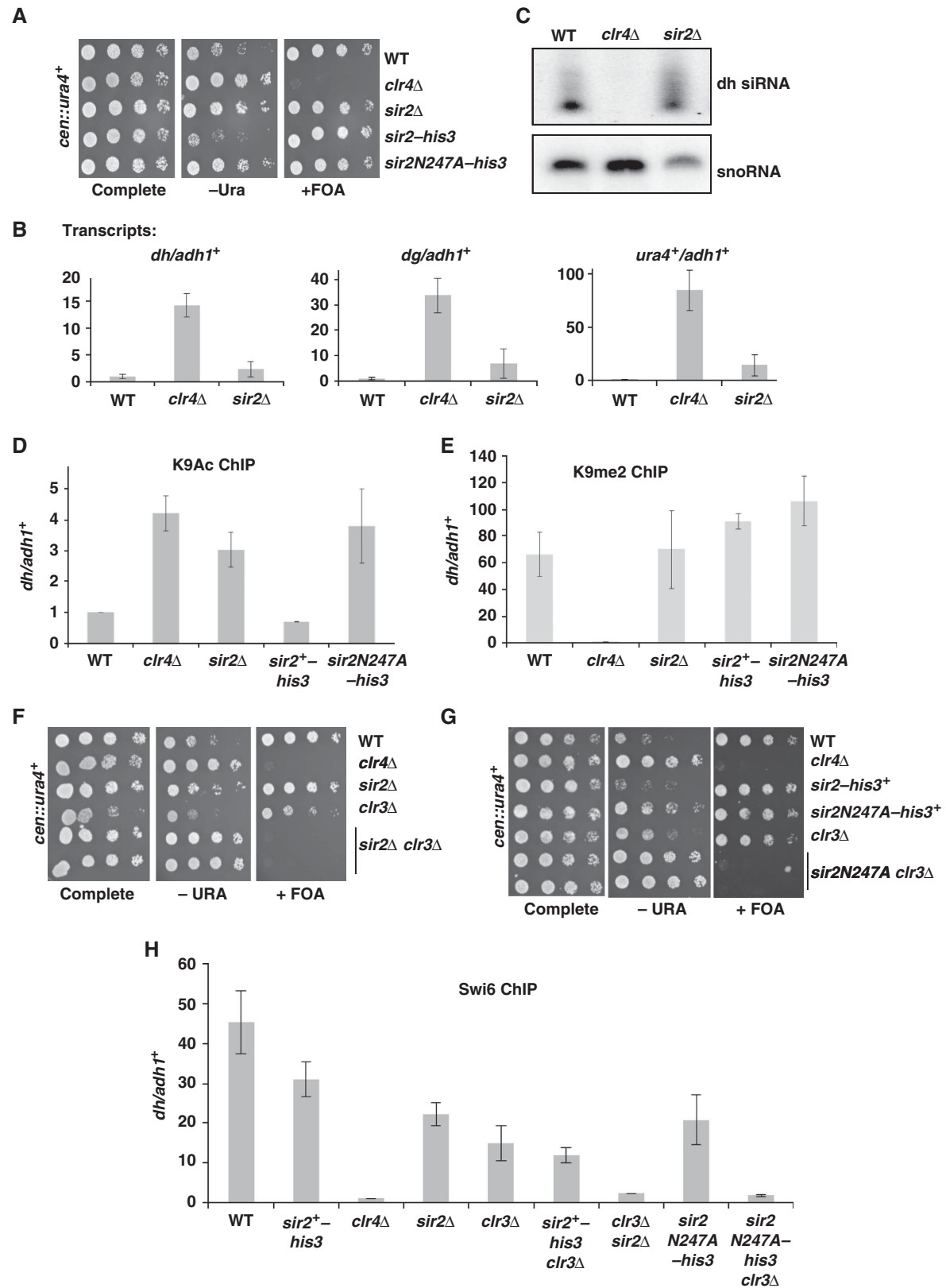

both transcription from the transgene and the endogenous centromeric repeats following reintegration of *clr4*$^+$ into *sir2Δclr4Δ*-deficient cells (Figure 3C). In contrast, in the controls, normal silencing of transcripts was reinstated on reintegration of *clr4*$^+$ into *clr4Δsir2*$^+$ backgrounds. We monitored *clr4*$^+$ transcripts following *clr4*$^+$ reintegration into the distinct backgrounds, and saw no evidence of reduction of *clr4*$^+$ transcript levels in *sir2Δ* backgrounds (Supplementary Figure S2A).

Frequently, failure to silence centromeric transcripts correlates with a defect in the RNAi pathway (Volpe *et al*, 2002). We analysed siRNAs derived from processing of centromeric *dh* transcripts by northern analysis (Figure 3D). Cells deficient in *clr4* largely lack siRNAs (Buhler *et al*, 2006; Halic and Moazed, 2010), but following reintegration of *clr4*$^+$, siRNAs accumulate to normal levels, demonstrating that the RNAi pathway is efficiently restored. Interestingly, centromeric siRNAs were also present following reintroduction of *clr4*$^+$ into *clr4Δsir2Δ* cells, suggesting that the failure to establish centromeric silencing in these cells is not caused by a defect in the initiation of the RNAi pathway. We confirmed that the defect is attributable to *sir2Δ*, since re-expression of *sir2*$^+$ in the *clr4*$^+$ reintegrant *clr4Δsir2Δ* cells fully compensated for the silencing defect (Supplementary Figure S2B and C).

We next assessed heterochromatin assembly by ChIP. We found that whereas robust H3K9me2 accumulates at centromeres on reintegration of *clr4*$^+$ into *clr4Δ* cells, reintegration of *clr4*$^+$ into *clr4Δsir2Δ* cells does not lead to enrichment of H3K9me2 on centromeric repeats above the background seen in *clr4Δ* strains (Figure 3E). In addition, levels of H3K9Ac and H3K14Ac on centromeric sequences remain high on reintegration of *clr4*$^+$ into *clr4Δsir2Δ* cells, similar to levels in *clr4Δ* cells (which are elevated above those seen in *sir2Δ* cells) (Figure 3F and G). In contrast, reintegration of *clr4*$^+$ into control *clr4Δ* cells permits deacetylation of K9Ac and K14Ac to levels found in WT cells. In addition, consistent with the low levels of centromeric H3K9me2, levels of Chp1 at centromeres on reintegration of *clr4*$^+$ into *sir2Δclr4Δ* cells remained low (Figure 3H). These results demonstrate that in cells lacking Sir2, there is a defect in the ability of Clr4 to direct the *de novo* assembly of centromeric heterochromatin.

We analysed the efficiency of chromosome segregation, and found that as expected for strains with defective centromeric heterochromatin, *sir2Δ clr4*$^+$ reintegrant strains accumulated lagging chromosomes during anaphase (Supplementary Figure S1). We also tested the stability of

the defect in heterochromatin initiation. The inability of *clr4Δsir2Δ* to *clr4*$^+$ reintegrant cells to establish centromeric heterochromatin was perpetuated over more than 100 generations, demonstrating that this is a stably inherited epigenetic defect (Supplementary Figure S3).

### Sir2 HDAC activity contributes to the establishment of centromeric heterochromatin

To test the role of Sir2's deacetylase activity in heterochromatin initiation, we generated the compound *clr4Δ sir2N247A* strain, and performed reintegration of *clr4*$^+$. Analysis of these strains demonstrated that the *sir2N247A* mutant showed an intermediate defect in the initiation of heterochromatin. Instead of a complete absence of growth on FOA, as seen in *clr4*$^+$ reintegrants into *clr4Δsir2Δ*, some *clr4Δsir2N247A clr4*$^+$ reintegrant cells grew on FOA (Figure 4A). Consistent with this, levels of transcripts arising from *dg* and *dh* regions of the centromere were reduced on reintegration of *clr4*$^+$ into *clr4Δsir2N247A* compared with levels seen in *clr4Δ sir2N247A*, but were still substantially elevated above those in the *sir2N247A* mutant background alone (Figure 4B). Monitoring the production of siRNAs in these strains revealed that siRNA production was efficiently restored on reintegration of *clr4*$^+$ into *sir2N247A* cells (Figure 4C). Together, these results suggest that the heterochromatin initiation is not caused by a defect in the RNAi pathway, and is largely dependent on the deacetylase function of Sir2.

How Sir2 promotes Clr4 function is not known. Sir2's H3K9 deacetylase activity could be necessary to provide the deacetylated H3K9 substrate to allow Clr4 to methylate H3K9. However, under steady-state conditions, we have demonstrated that cells lacking Sir2 show elevation of H3K9Ac at centromeres, without impacting H3K9 methylation (Figure 2D and E). We therefore questioned whether H3K9Ac is the critical target of Sir2, or whether Sir2 targets other acetyl moieties to promote heterochromatin initiation.

### Substrate specificity of Sir2 in vitro

We purified recombinant GST fusion proteins of WT and N247A mutant Sir2 (Figure 5A), and tested these proteins on different acetylated or non-acetylated peptides in deacetylase assays. Using a fluorogenic substrate (Figure 5B), we demonstrated that GST–Sir2 is a catalytically active lysine deacetylase. In contrast, the GST–sir2N247A mutant showed no deacetylase activity above background in these assays. Consistent with prior studies (Bitterman *et al*, 2002; Avalos

**Figure 2** Sir2 functions redundantly with Clr3 for centromeric heterochromatin maintenance. (**A**) Deletion of *sir2* or *sir2N247A* does not impact centromeric transgene silencing. Serial dilutions of WT, *clr4Δ*, *sir2Δ*, and *sir2N247A* yeast strains bearing the *cen::ura4*$^+$ transgene were assayed for growth on nonselective (complete) medium, as well as selective medium lacking uracil (−URA), and complete medium additionally containing +FOA. (**B**) Centromeric transcripts do not appreciably accumulate in *sir2* mutants. Centromeric *dh, dg,* and *ura4*$^+$ mRNA transcripts were quantified by quantitative real-time PCR amplification of cDNA, and normalized to transcript levels of the *adh1*$^+$ control. Graphic data present the average of two distinct experimental replicates and error bars depict s.e.m. (**C**) siRNA production is unaffected in *sir2* mutants. Centromeric *dh* siRNA production was evaluated by northern blot relative to snoRNA controls in WT, *clr4Δ*, and *sir2Δ* mutants. (**D**) H3K9 acetylation at centromeres is increased in *sir2* mutants, while H3K9 dimethylation, (**E**) is unaffected. ChIP experiments were performed using antibodies against the acetylated or dimethylated H3K9 epitope. Specific enrichment of immunoprecipitated centromeric *dh* sequences was determined relative to the *adh1*$^+$ euchromatic locus by real-time PCR. Graphical data present the average of two biological replicates, with error bars depicting the s.e.m. (**F**) *sir2Δ* and (**G**) *sir2N247A* mutants function redundantly with Clr3 to maintain centromeric transgene silencing. Serial dilutions of yeast strains bearing the *cen::ura4*$^+$ transgene were assayed for growth on nonselective (complete) medium, as well as selective medium lacking uracil (−URA), and complete medium additionally containing +FOA. (**H**) Sir2 and Clr3 function redundantly to maintain centromeric heterochromatin. Swi6 recruitment to centromeres is lost in *sir2 clr3* compound mutants. ChIP experiments were conducted using antibodies specific for Swi6, and enrichment of centromeric *dh* sequences relative to *adh1*$^+$ was quantified by real-time PCR and normalized to input DNA. Data represent the average of two biological replicates, with error bars depicting the s.e.m.

*et al*, 2005), we further demonstrated that the deacetylase activity of GST–Sir2 is blocked by addition of nicotinamide, a recognized sirtuin inhibitor.

Sirtuins are $NAD^+$-dependent deacetylases. Lysine deacetylation by Sir2 is coupled to the obligate hydrolysis of the $NAD^+$ cofactor and, conversely, $NAD^+$ hydrolysis by Sir2

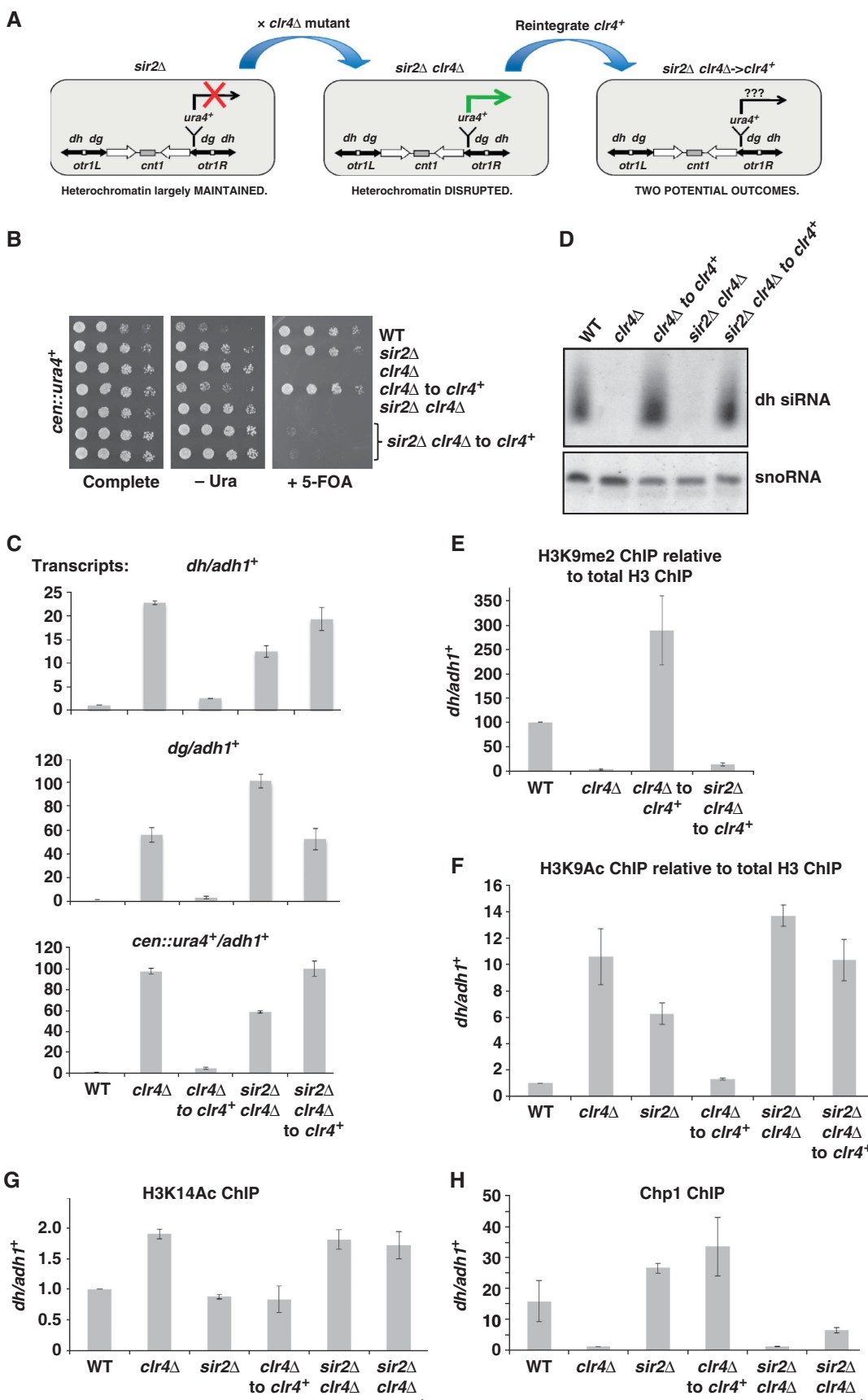

requires coupled lysine deacetylation. Quantitative analysis of radiolabelled $^{32}$P NAD$^+$ hydrolysis in the presence of acetylated histone or peptide substrates has therefore been reported to provide a measure of the degree to which putative

sirtuin substrates serve as deacetylation targets (Tanny and Moazed, 2001). By evaluating coupled $^{32}$P NAD$^+$ hydrolysis, we demonstrated that GST–Sir2, but not GST–sir2N247A, promotes lysine deacetylation and thus $^{32}$P NAD$^+$ hydrolysis

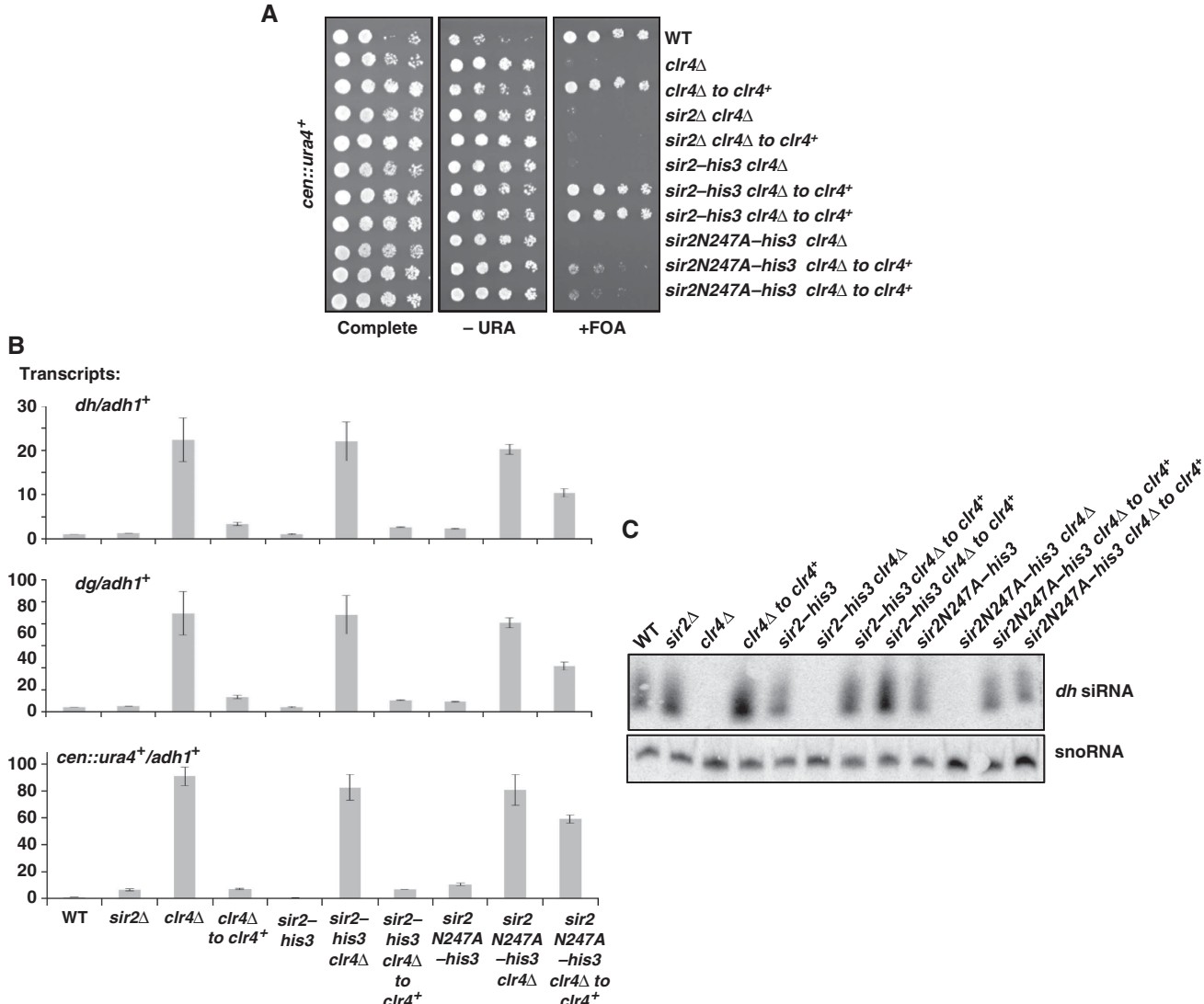

**Figure 4** The *sir2N247A* mutant exhibits defects in *de novo* assembly of heterochromatin at centromeres. (**A**) *sir2N247A* cells show some defect in establishment of centromeric silencing. Serial dilution assays were performed to monitor silencing of *cen::ura4* transgene. (**B**) *De novo* silencing defects observed in yeast expressing *sir2N47A* reflect defective transcriptional silencing at centromeres. Centromeric *dh, dg,* and *ura4$^+$* mRNA transcripts were quantified by quantitative real-time PCR and normalized to levels of the euchromatic *adh1$^+$* control. Data represent the average of two experimental replicates, including a total of four biological replicates from two distinct *sir2N247A-his3 clr4Δ to clr4$^+$* reintegrant strains. (**C**) *sir2N247A* mutation does not disrupt siRNA production during maintenance or establishment. Centromeric *dh* siRNA production was evaluated by northern blot relative to that of a snoRNA control in each of the cell backgrounds indicated.

**Figure 3** Sir2 is required for *de novo* silencing and establishment of Clr4-dependent H3K9 methylation at centromeres. (**A**) Schematic of assay used to evaluate the requirement for *sir2$^+$* in promoting *de novo* clr4$^+$-dependent silencing activity at centromeres. (**B**) Sir2 is required for *de novo* centromeric silencing upon reintroduction of the *clr4$^+$* methyltransferase. Serial dilutions of yeast bearing the *cen::ura4$^+$* transgene were assayed for growth on complete medium, as well medium lacking uracil (−URA), and complete medium containing +FOA. (**C**) Reintroduction of the *clr4$^+$* methyltransferase in sir2Δ clr4Δ cells reveals defective establishment of transcriptional silencing at centromeres. Centromeric *dh, dg,* and *ura4$^+$* mRNA transcripts were quantified by quantitative real-time PCR amplification of random primed cDNA and normalized to transcript levels for the euchromatic *adh1$^+$* control. Experimental data present the average of two experimental replicates. (**D**) siRNA production occurs in cells defective for *de novo* centromeric silencing. Centromeric *dh* siRNA production was evaluated by northern blot relative to snoRNA controls. (**E–H**) Changes in chromatin structure underlie transcriptional silencing defects in cells defective for *de novo* centromeric silencing. ChIP experiments were performed using antibodies specific for dimethylated (**E**) or acetylated H3K9 (**F**), acetylated H3K14 (**G**) and Chp1 (**H**), and data represent the average of two experimental replicates, including a total of four biological replicates from two distinct sir2Δ clr4Δ to *clr4$^+$* reintegrant strains.

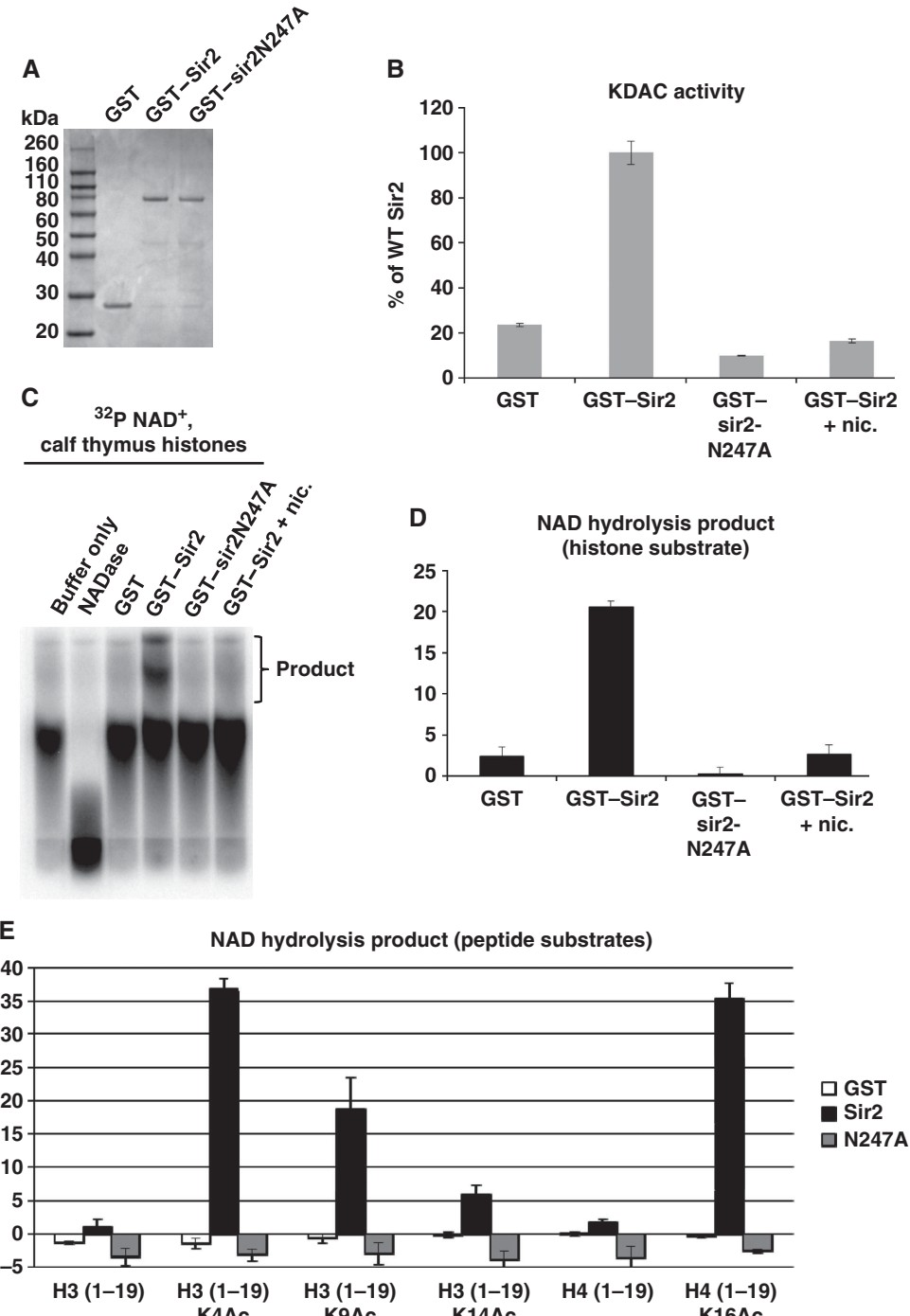

**Figure 5** Sir2, but not the catalytic point mutant sir2N247, is catalytically active *in vitro*, and exhibits preference for deacetylation of H3K4Ac and H4K16Ac peptides. (**A**) Affinity purified recombinant GST–Sir2 and GST–sir2N247A are stable proteins. Equal amounts ($\sim 1\,\mu$g) of GST, GST–Sir2, and GST–sir2N247A were resolved by SDS–PAGE and stained using Coomassie brilliant blue. (**B**) GST–Sir2, but not GST–sir2N247A, is catalytically active and sensitive to inhibition by nicotinamide. Catalytic activity of GST fusion proteins was evaluated by fluorogenic deacetylation assay. GST–Sir2 activity was also evaluated in the presence 2 mM nicotinamide, a sirtuin inhibitor. (**C**, **D**) GST–Sir2, but not GST–sir2N247A, promotes $^{32}$P NAD$^+$ cofactor hydrolysis in the presence of calf thymus histones. (**C**) The ability of GST fusions to promote hydrolysis of $^{32}$P NAD$^+$ was appraised in the presence of acetylated calf thymus histones by TLC and autoradiography. GST–Sir2 activity was also evaluated in the presence 5 mM nicotinamide. (**D**) Graphical representation of data from panel C, one of the two experimental replicates, following analysis by quantitative densitometry. (**E**) Sir2 exhibits preference for deacetylation of H3KAc and H4K16Ac peptides, but also deacetylates H3K9Ac and H3K14Ac peptides. Evaluation of $^{32}$P NAD$^+$ hydrolysis products was performed following the coupled NAD$^+$ hydrolysis/lysine deacetylation reaction, in the presence of differentially acetylated or unacetylated peptides derived from the N-terminal 19 amino acids of *Schizosaccharomyces pombe* histones H3 or H4. TLC autoradiographs were evaluated by computational densitometry. Graphs represent the mean of two experimental replicates, for which representative autoradiographs are presented in Supplementary Figure S2A–C.

in the presence of acetylated histones from calf thymus, and that this activity is abrogated in the presence of nicotinamide (Figures 5C and D).

To explore the specificity of Sir2's deacetylase activity in greater detail, we generated histone tail peptides for histone H3$_{(aa\ 1-19)}$ and H4$_{(aa\ 1-19)}$ that bear different acetyl marks.

Specifically, we made peptides bearing H4K16Ac (an extensively studied target of budding yeast Sir2) (Tanny and Moazed, 2001), H3K9Ac (previously reported as a substrate for fission yeast Sir2; Shankaranarayana *et al*, 2003), H3K14Ac, and H3K4Ac—a newly reported acetyl mark on H3 that appears to be regulated by Sir2 *in vivo* (Garcia *et al*, 2007; Xhemalce and Kouzarides, 2010). As can be seen in Figure 5E and Supplementary Figure S4, Sir2 displayed robust NAD$^+$-dependent deacetylase activity when supplied with the canonical sirtuin target H4K16Ac, while sir2N247A lacked such activity. Similar to H4K16Ac, H3K4Ac provided an excellent substrate for Sir2 deacetylase activity. Both H4K16Ac and H3K4Ac appear to be better targets than H3K9Ac. Interestingly, we show that Sir2 can directly deacetylate H3K14Ac. Previously, it had been shown that the total cellular level of H3K14Ac was not altered in histones isolated from *sir2∆* cells but that levels of H3K4Ac were increased (Xhemalce and Kouzarides, 2010). Increased levels of H3K14Ac have been documented at heterochromatic loci in *sir2∆* cells and were attributed to loss of heterochromatin in these cells (Shankaranarayana *et al*, 2003). We asked whether H4K16Ac levels were affected by *sir2$^+$* deletion, and found a small but reproducible increase in total cellular H4K16Ac in *sir2∆* cells (Supplementary Figure S4D). Here, we demonstrate that fission yeast Sir2 can directly deacetylate both H3K14Ac and H3K4Ac *in vitro*.

### H3K4 and the canonical Sir2 target H4K16 do not contribute to heterochromatin initiation

Our *in vitro* assays suggested that H3K4Ac, H4K16Ac, H3K9Ac, and H3K14Ac are all substrates for Sir2 deacetylase activity. We therefore attempted to assess the *in vivo* relevance of these histone targets for centromeric heterochromatin initiation. To do this, we used fission yeast bearing lysine mutations in analogous positions of the H3 and H4 histone tails.

Fission yeast encode three copies each of histones H3 and H4, but strains have been developed that retain just one copy of histones H3 and H4, which appear relatively WT (Mellone *et al*, 2003). Several mutants have been generated in these strains in which lysine residues have been mutated to mimic the charge of an acetylated lysine (K to G and A or Q) as a proxy for loss of specific HDAC activity on a particular residue.

Mutation of H4K16 to G has been shown to have no impact on centromeric heterochromatin maintenance in fission yeast (Mellone *et al*, 2003). H3K9A and H3K14A mutants, however, exhibit loss of heterochromatin ((Mellone *et al*, 2003) and Figures 6A and B) and so cannot be tested for effects on heterochromatin initiation via the *clr4$^+$* withdrawal/ reintegration assay. To our knowledge, no mutants in fission yeast H3K4 had been reported apart from a H3K4 to R mutant, which maintains this residue's basic charge (Xhemalce and Kouzarides, 2010). We therefore generated amino-acid substitutions at K4 to mimic the lack of charge of an acetylated residue at this position. We found that H3K4G mutants are not viable when serving as the only source of histone H3 expressed in cells. H3K4A mutants are viable and show little defect in centromeric heterochromatin maintenance (Figure 6A and B). H3K4Q mutant strains are also viable, and maintain silencing of centromeric *dg*

transcripts but exhibit elevated levels of centromeric *dh* transcripts (Supplementary Figure S5A and B).

To address the role of these residues in centromeric heterochromatin initiation, we made compound mutant strains with *clr4∆*, and performed *clr4$^+$* reintegration experiments in *clr4∆* H4K16G, *clr4∆* H3K4A, and *clr4∆* H3K4Q mutants, alongside *clr4∆* strains bearing single-copy WT H3 and H4 genes. Analysis of centromeric transcripts demonstrated that on reintegration of *clr4$^+$* into single-copy WT histone H3 and H4 *clr4∆* strains (Figure 6C and D), both *dg* and *dh* transcripts are reduced to background levels. Somewhat surprisingly, we found that *dg* and *dh* transcripts were also reduced to near background levels on reintegration of *clr4$^+$* into the *clr4∆* H3K4A strain (Figure 6C and D). Similar results were obtained for *dg* transcripts on reintegration of *clr4$^+$* into the *clr4∆* H3K4Q strains (Supplementary Figure S5C). When *clr4$^+$* was reintegrated into the *clr4∆* H4K16G strain (Figure 6E and F), *dg* and *dh* transcripts were also downregulated to near WT levels.

Together, these data demonstrate that the substitutions at two of Sir2's targets: H3K4 and H4K16, with residues that mimic the lack of charge of an acetylated residue, do not impede the ability of Clr4 to initiate assembly of centromeric heterochromatin. This leads us to conclude that either another acetylated histone target of Sir2, such as H3K9Ac or H3K14Ac, a combination of acetylated histone targets, or perhaps an acetylated non-histone chromatin target contributes to the centromeric heterochromatin establishment defect seen in cells lacking Sir2 activity.

### H3K14 is critical for Clr4 recruitment to centromeres

Testing the role of H3K14 and H3K9 during centromeric heterochromatin assembly is difficult, since mutations at K9 or K14 of histone H3 cause disruption of centromeric heterochromatin. We did attempt to address whether H3K14A mutation blocked heterochromatin initiation (Supplementary Figure S6), but interpretation was hampered by the high levels of centromeric transcripts that accumulate under normal growth of this strain. That mutation of H3K9 phenocopies loss of *clr4* function is not unexpected, since K9 is the target for methylation by Clr4. However, it is presently unclear why mutation of K14 should disrupt centromeric heterochromatin. *In vitro* studies have demonstrated that a peptide bearing H3K14Ac modification can be efficiently methylated on K9 by Clr4 or its human homologue Suv39h1 (Nakayama *et al*, 2001; Rea *et al*, 2000), and binding of H3K9me2 'reader' proteins, such as Swi6, is not altered by H3K14Ac on peptide substrates (Yamada *et al*, 2005). We therefore questioned whether *in vivo* recruitment of Clr4 to centromeres might be impacted by acetylation of K14 within the histone H3 tail. We used the H3K14A acetyl mimic strain to address this question.

Compound mutant strains were generated from strains bearing single-copy mutant H3 (K9A or K14A) and *clr4∆*, and were transformed with a plasmid expressing *3xFlag-clr4$^+$* under control of its native regulatory sequences. This episomal Flag–Clr4 fully complements for *clr4* function at centromeres (Supplementary Figure S7A). Western analyses demonstrated that expression of Flag–Clr4 protein was similar between these strains (Figure 6G). ChIP experiments revealed that whereas Flag–Clr4 was efficiently recruited to

centromeres in *clr4Δ* strains expressing single-copy WT H3, Flag–Clr4 association with centromeres was reduced to background levels in cells expressing H3K9A or H3K14A

(Figure 6H). Consistent with this result, Swi6 was not recruited to centromeres in cells expressing H3K14A (Supplementary Figure S7B).

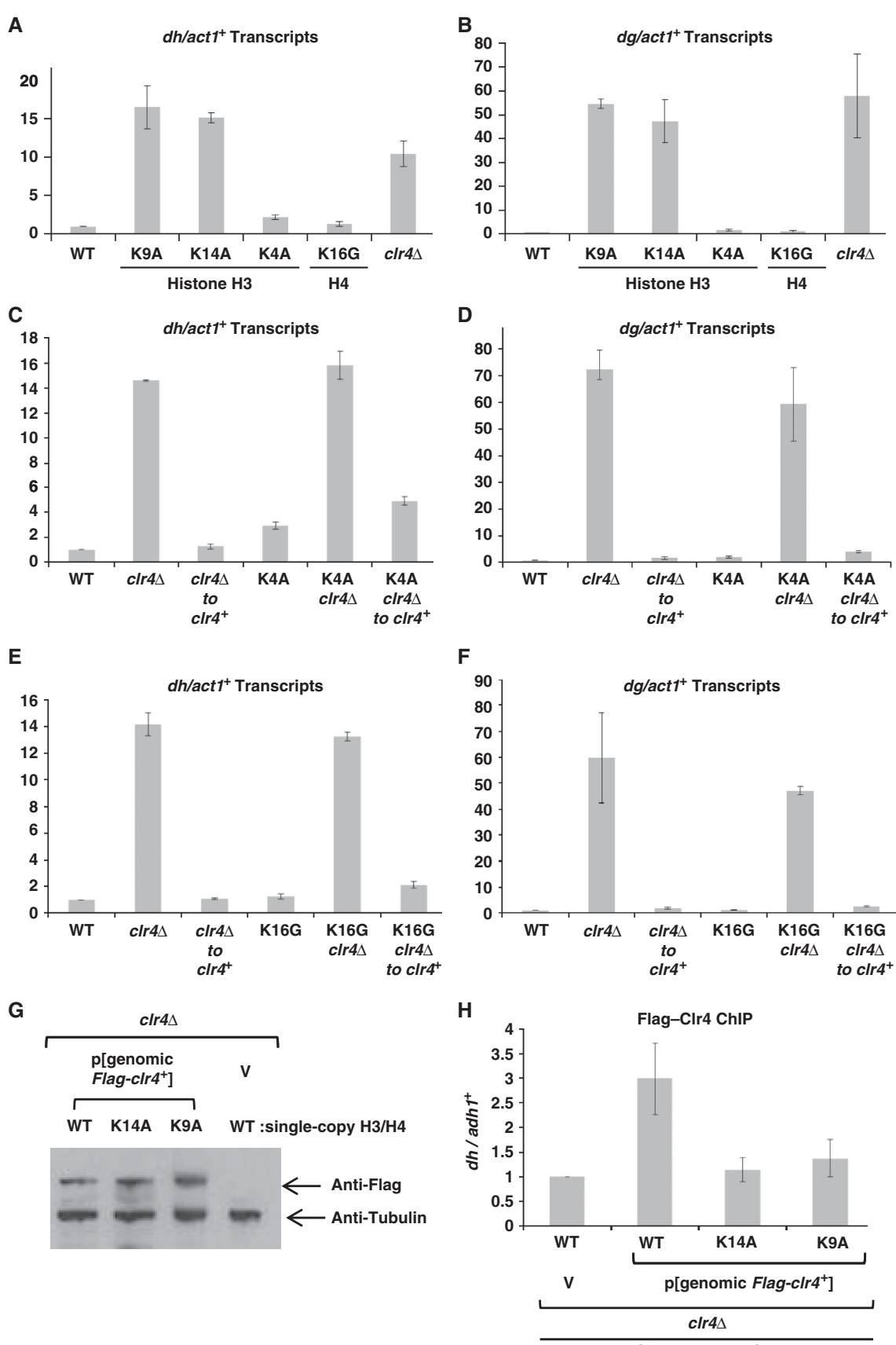

These experiments suggest that the presence of amino-acid substitutions at K9 or K14 of histone H3 that mimic the charge of an acetylated lysine cause a defect in the recruitment of Clr4 to centromeres. Since Sir2 can deacetylate K9 and K14 of H3, we might infer that deacetylation of these residues by Sir2 is necessary for Clr4 recruitment to centromeres to initiate heterochromatin assembly. Alternatively, the lack of recruitment to centromeres in the H3K9A and H3K14A mutant strains could be an indirect effect of the high levels of transcription at centromeres in these backgrounds. Counter to this argument, previously we had analysed the ability of Clr4 to mediate H3K9 methylation when overexpressed in mutant backgrounds that exhibit high levels of transcriptional activity at centromeres. We found that in cells lacking *clr4* and *dcr1* or *clr4* and *ago1*, low levels of H3K9 methylation could be detected on centromeric repeats following reintroduction of episomal genomic *clr4+* (Shanker *et al*, 2010). To further probe the role of Sir2 in heterochromatin

assembly, we asked whether Sir2 was necessary for accumulation of this methyl mark in RNAi-deficient cells.

### Sir2 is required for the de novo deposition of H3K9me2 in RNAi-deficient cells

To test whether Sir2 contributes to the ability of Clr4 to methylate H3 in RNAi-deficient backgrounds, we generated strains triply deleted for *clr4, dcr1*, and *sir2*. We then expressed *clr4+* from an episomal vector, and monitored by ChIP whether H3K9me2 and Swi6 could accumulate at centromeres in these strains (Figure 7A). Whereas significant accumulation of H3K9me2 was detected on overexpression of *clr4+* in *clr4Δ* or even *clr4Δdcr1Δ* cells, levels of H3K9me2 in *clr4+*-transformed *clr4Δdcr1Δsir2Δ* cells did not differ from *clr4Δ* cells transformed with empty vector.

To confirm and extend these results, we monitored enrichment of Swi6 on centromeric repeats in these strain backgrounds (Figure 7B). Similar to the results obtained for

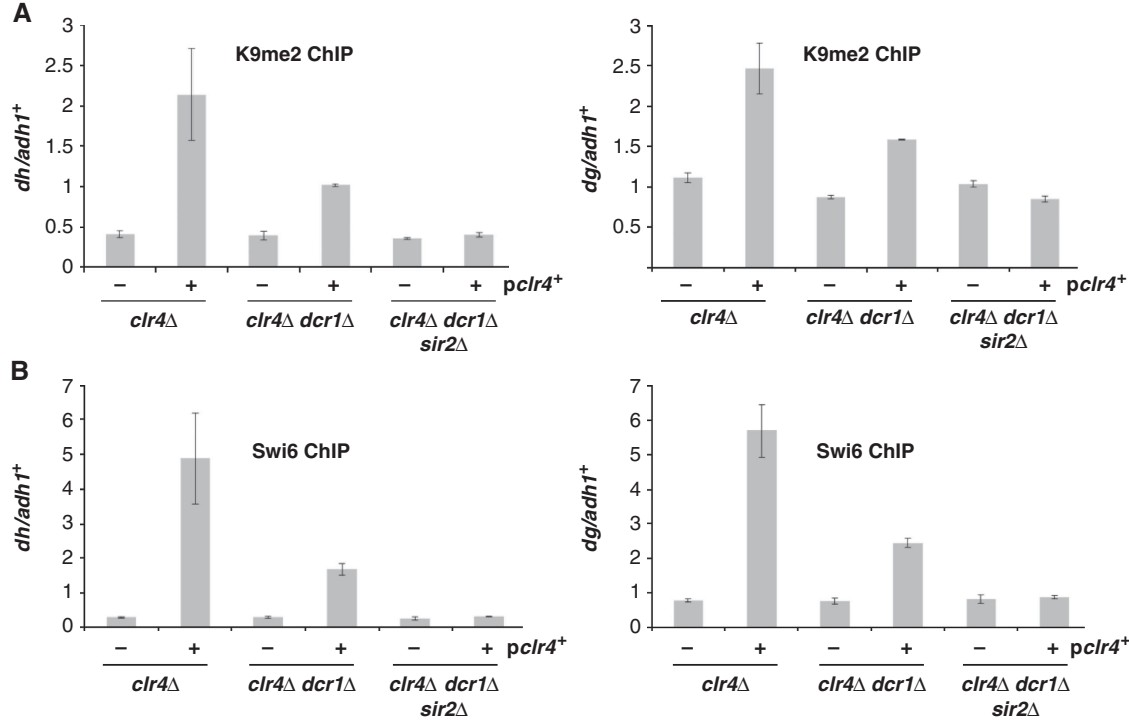

**Figure 7** *De novo* H3K9 methylation and Swi6 recruitment in RNAi-deficient backgrounds is dependent on Sir2. (**A**) Sir2 is required for centromeric H3K9 methylation upon *clr4+* expression in RNAi-defective cells. Quantitative real-time PCR analysis of ChIP experiments performed with anti-H3K9me2 antibody, monitoring relative enrichment of centromeric *dg* and *dh* sequences over *adh1+* euchromatic locus following episomal *clr4+* expression in *clr4Δ*, *clr4Δdcr1Δ*, and *clr4Δdcr1Δsir2Δ* strains. Results are s.e.m. for duplicate independent ChIP experiments, with inclusion of duplicate data for two biological replicates for *clr4Δdcr1Δsir2Δ* strains. (**B**) Sir2 is required for centromeric Swi6 recruitment upon *clr4+* expression in RNAi-defective cells. Swi6 ChIP experiments performed on strains outlined in (**A**), assessing relative enrichment at centromeric *dg* and *dh* sequences relative to *adh1+* control.

**Figure 6** H3K4 and H4K16 do not contribute to *de novo* centromeric silencing. (**A, B**) The integrity of H3K9 and H3K14, but not H3K4 or H4K16, is critical to maintenance of centromeric silencing. Accumulation of centromeric transcripts from the *dh* (A) and *dg* (B) repeats was evaluated by quantitative real-time PCR of cDNA derived from yeast expressing sole copies of histone H3 and histone H4 with the mutations indicated. Euchromatic *act1+* transcripts serve as a normalization control. Neither H3K4 (**C, D**) nor H4K16 (**E, F**) are critical for *de novo* centromeric silencing mediated by *clr4+*. Experimental data present the average of two experimental replicates, including a total of four biological replicates from two distinct *H3K4A clr4Δ* to *clr4+* and *H4K16G clr4Δ* to *clr4+* reintegrant strains. (**G**) Flag–Clr4 is equivalently expressed in cells bearing mutations in histone H3. Western analysis of *clr4Δ* cells bearing different histone H3 mutations and transformed with episomal Flag–Clr4 expression vector demonstrates equivalent Flag–Clr4 expression compared with tubulin loading control. (**H**) Flag–Clr4 is efficiently recruited to centromeres, but fails to localize in H3K9A and H3K14A mutant cells. Q–PCR analysis of ChIP for Flag–Clr4 recruitment to centromeric *dh* sequences compared with *adh1+* euchromatic control. Data averaged from five experimental replicates, with s.e.m. shown.

H3K9me2, we did not detect Swi6 at centromeres on overexpression of $clr4^+$ in $clr4\Delta dcr1\Delta sir2\Delta$ cells, whereas Swi6 was enriched at centromeres in $clr4\Delta dcr1\Delta$ cells overexpressing $clr4^+$. These data reinforce that Clr4 activity and the H3K9 methylation seen in RNAi-defective cells is dependent on Sir2.

## Discussion

The assembly of heterochromatin relies on a complex interplay between factors that are important for the establishment of heterochromatin and for its maintenance. Here, we have uncovered novel roles for the Sir2 histone deacetylase in the establishment of heterochromatin in fission yeast. Our studies reveal that Sir2 is necessary for promoting Clr4-mediated H3K9 methyltransferase activity and establishment of transcriptional silencing at centromeres. We further demonstrate that *de novo* recruitment of H3K9 methylation and transcriptional silencing mediated by Sir2 is highly dependent on the enzyme's deacetylase activity. This is supported by our observation that at centromeres, *de novo* silencing upon $clr4^+$ reintroduction is largely lost in sir2N247A mutant backgrounds.

How does Sir2 promote Clr4 activity? We hypothesize that Sir2 may act to create a nucleosomal environment that is permissive to Clr4 binding and activity. Here, we provide the first direct evidence that H3K4Ac and K14Ac are indeed *bona fide* targets of the fission yeast Sir2 activity. We further demonstrate that the enzyme preferentially deacetylates H3K4Ac and H4K16Ac, and to a lesser degree H3K9Ac and H3K14Ac peptides *in vitro*. Surprisingly, despite the apparent *in vitro* specificity of Sir2 deacetylase activity towards H3K4Ac and H4K16Ac, these substrates do not appear to be critical to the establishment of centromeric heterochromatin. Indeed, H3K4A and H4K16G mutants intended to mimic the charge of constitutively acetylated residues, as might be expected upon *sir2* deletion or catalytic mutation, were found to have little impact on either maintenance or establishment of centromeric silencing.

However, we cannot rule out contributions to our centromeric heterochromatin establishment phenotype from two other histone targets of Sir2: namely, H3K9Ac and H3K14Ac. H3K9A and H3K14A mutants are known to profoundly affect centromeric silencing (Mellone *et al*, 2003). Loss of heterochromatin is expected in cells bearing H3K9A mutations, since H3K9 is the critical target of Clr4 histone methyltransferase activity (Rea *et al*, 2000). It is unclear, however, why mutation of H3K14 should cause heterochromatin loss similar to that of cells lacking *clr4*, with commensurate loss of chromatin association by Swi6 (Mellone *et al*, 2003).

*In vitro* studies have demonstrated that recombinant Clr4 can methylate H3K9 even on peptides bearing H3K14Ac, suggesting that the Clr4 H3K9 methyltransferase activity is not abrogated by charge differences at H3K14 (Nakayama *et al*, 2001). In addition, the interaction of the Swi6 HP1-like chromodomain protein with the histone H3K9me2 tail is not affected by K14Ac *in vitro* (Yamada *et al*, 2005). However, we note that these experiments are performed under conditions of enzyme or effector excess, and use histone tail peptides rather than nucleosomal substrates. To our knowledge, the impact of H3K9 or H3K14 mutation on Clr4 association with nucleosomal targets has not been previously evaluated.

Because the epitope for commercially available H3K9me2 antibodies overlies the K14 residue, we could not directly test whether Clr4 promoted H3K9 methylation at centromeres in H3K14A mutants by ChIP experiments. Instead, we monitored the association of Clr4 with centromeres in cells bearing histone H3 mutant chromatin. We found that in H3K14A or H3K9A mutant cells, Clr4 association with centromeres was lost. This failure to recruit Clr4 to chromatin would cause loss of H3K9me2 and mislocalization of H3K9me2-binding proteins such as Swi6 (shown in Supplementary Figure S7B) and could explain the profound loss of centromeric heterochromatin in H3K14A mutant cells.

Why do H3K9A and H3K14A mutations result in loss of localization of Clr4? One clear possibility is that in these mutant backgrounds there are high levels of transcription through the centromeric repeats, which may perturb the ability of Clr4 to be recruited. However, our experiments comparing Clr4 function in cells bearing multiple mutations in components required for centromeric heterochromatin assembly reveal differential Clr4 function in cells that all express high levels of centromeric transcripts (Figure 7). These data suggest that it is not the level of transcription *per se* that is blocking Clr4 function, but that the Clr4 recruitment defect is specifically linked to loss of Sir2 function. Clr4 is a member of the Clr–C complex, and *in vivo* the function of Clr4 depends on all components of Clr–C (Hong *et al*, 2005; Horn *et al*, 2005; Jia *et al*, 2005; Li *et al*, 2005; Thon *et al*, 2005). Little is known about how this complex associates with chromatin, but it is possible that H3K14 acetylation or H3K14A mutation blocks recruitment of another Clr–C component to chromatin, and thereby indirectly blocks localization of Clr4. In agreement with this, H3K14R mutant cells also display disruption of heterochromatin, and a defect in Clr4 recruitment to centromeres (Mellone *et al*, 2003, Supplementary Figure S7C and D).

For H3K9A mutants, another possible explanation stems from loss of identity of the H3K9 residue itself. The N-terminal chromodomain of Clr4 has been shown to bind H3K9me2, resulting in a positive feedforward mechanism to amplify the H3K9me2 signal (Zhang *et al*, 2008). H3K9me2 appears critical for recruitment of Clr4, since mutation of Clr4's histone methyltransferase domain to cause loss of centromeric H3K9me2 leads to a defect in Clr4 recruitment (Supplementary Figure S8).

Several histone deacetylases in fission yeast can deacetylate H3K14. These include Clr3, which is thought to be specific for H3K14 deacetylation, and Clr6, which is a broad spectrum deacetylase (Bjerling *et al*, 2002). One might predict that strains bearing mutations in these enzymes would show similar phenotypes to *sir2* mutants. Indeed, similar to *sir2*Δ, cells lacking Clr3 and mutant for Clr6 show a profound loss of silencing of subtelomeric sequences (Hansen *et al*, 2006) and at centromeres, a redundant role for Clr3 in maintenance of H3K9me2 heterochromatin is revealed in cells additionally mutated for the RNAi pathway (Yamada *et al*, 2005). Sir2 HDAC activity is clearly required for Clr4 function to maintain heterochromatin within subtelomeric domains, but loss of *sir2* has little impact on the maintenance of centromeric heterochromatin. We note that the RNAi pathway plays a critical role in the maintenance of centromeric heterochromatin (Volpe *et al*, 2002), but is relatively unimportant for maintenance of subtelomeric

heterochromatin (Petrie et al, 2005). It is possible, and has very recently been demonstrated (Buscaino et al, 2013) that, similar to Clr3 (Yamada et al, 2005), Sir2 functions redundantly with RNAi in centromeric heterochromatin maintenance. However, Sir2's role in promotion of Clr4 activity is revealed at subtelomeres where heterochromatin is maintained through an RNAi-independent pathway. Indeed, we present here that Sir2 and Clr3 play redundant roles in the maintenance of centromeric heterochromatin (Figure 3), as has recently been reported (Buscaino et al 2013). Additionally, we demonstrate an important dependence on Sir2 for Clr4 function during the RNAi-independent establishment of heterochromatin at centromeres (Figure 7; Partridge et al, 2007; Shanker et al, 2010). We note that Sir2 enzymatic activity has recently been reported as sufficient and necessary for maintenance of heterochromatin in RNAi-deficient yeast (Buscaino et al, 2013), although this study did not reveal a dependence on Sir2 for heterochromatin initiation.

Why does sir2N247A promote limited de novo silencing of centromeres upon clr4+ reintroduction? One possibility is that the enzyme retains a low level of histone deacetylase activity. Though our data clearly demonstrate that GST–sir2N247A is catalytically inactive in vitro, it is conceivable that this enzyme may have very low levels of activity that suffice to promote limited H3K9 and H3K14 deacetylation, and thereby limited clr4+ activity and recruitment. We speculate that sir2N247A may serve in a structural capacity that is sufficient to nucleate low-level silencing at centromeres. An intriguing possibility is that sir2N247A may serve as a scaffold that functions in concert with other histone deacetylases, such as Clr3, which we have demonstrated to act redundantly with sir2N247A (Figure 3). A wholly distinct possibility is that Sir2 physically interacts with Clr4 to directly recruit its activity to centromeres and other loci, and that this physical association is required for Clr4 function. In mammals, the Sir2 homologue, SirT1, has been shown to deacetylate the homologue of Clr4, Suv39h1, to promote its methyltransferase activity (Vaquero et al, 2007). In addition, binding of SirT1 to Suv39h1 masks a site of ubiquitination on Suv39h1, such that in cells lacking SirT1 protein, Suv39h1 protein levels are decreased (Bosch-Presegue et al, 2011). However, we have conclusively demonstrated that loss of Sir2 protein does not impact the transcription of clr4+ or steady-state levels of expression of endogenous Clr4 protein (Supplementary Figure S9). While it remains possible that Sir2 deacetylates Clr4 to promote its activity, the deacetylation site is not conserved between Suv39h1 and Clr4, and experiments to test this possibility lie outside the scope of the current work.

How Sir2 is itself recruited to centromeres remains an open question. We have demonstrated previously that the initial recruitment of Clr4 activity to centromeres occurs independent of the RNAi pathway (Partridge et al, 2007; Shanker et al, 2010). By extension, centromeric Sir2 recruitment may also occur independently of RNAi. Indeed, our experiments demonstrating a dependence on Sir2 for Clr4 function at centromeres in RNAi-deficient yeast (Figure 7) would strongly argue that Sir2 localization occurs independently of RNAi. This is supported by recent experiments revealing that Sir2 localization to centromeres is independent of both Clr4 and RNAi (Buscaino et al, 2013). In budding yeast, the initial

recruitment of Sir2 to telomeres is mediated via association of the SIR complex with silencing factors that bind to specific DNA sequences within telomeric repeats such as Rap1p (Hickman et al, 2011). Given the recent findings concerning the role of transcription factors in the assembly of H3K9me2 chromatin at mammalian centromeres (Bulut-Karslioglu et al, 2012), we think it likely that Sir2 is recruited via DNA-binding factors to initiate the assembly of centromeric heterochromatin in fission yeast.

We note that while this paper was in revision, an elegant paper was published in EMBO J from the Allshire group that has identified regions within centromeric dg sequences that contribute to Sir2-dependent H3 K9 methylation (Buscaino et al, 2013). In summary, the combined work from the two papers has helped to reveal the mechanism (in terms of DNA sequence requirements and deacetylation targets) underlying the critical role for Sir2 in establishing and maintaining centromeric heterochromatin. Future studies will reveal how Sir2 is targeted to particular loci to establish heterochromatin, and whether similar strategies are employed to recruit Sir2 to mammalian centromeres.

# Materials and methods

### Strain generation
Strains used in this study are listed in Supplementary Table S1. Please see Supplementary data for additional information.

### Plasmid construction
Please see Supplementary data.

### Transcript analyses
These were performed essentially as described (Partridge et al, 2007; Debeauchamp et al, 2008). Real-time primers for act1 (JPO-2000, 2001) are mbp86 and 87 from (Buhler et al, 2007).

### siRNA analyses
siRNA preparation was as performed previously, and siRNA from dh repeats and snoR69 as loading control were detected by hybridization (Partridge et al, 2007).

### Cell growth analyses
These were performed as described previously (Debeauchamp et al, 2008). To assess the maintenance of silencing in strains transformed with his3+ plasmids, cells were grown in PMG–his medium and were plated on minimal medium lacking histidine (PMG – his), medium lacking histidine and uracil (PMG – his – ura), or minimal medium lacking histidine and supplemented with 2 g FOA per litre (PMG – his + FOA). A similar approach was taken for strains bearing LEU2+ vectors.

### Western analyses
Antibodies used: 'Anti-TAP': HRP conjugated rabbit IgG (Jackson ImmunoResearch Laboratories #011-030-00), Flag: M2 monoclonal antibody (Sigma #F3165), Tubulin: TAT1 monoclonal from K. Gull. H4K16Ac: Rabbit polyclonal IgG (Active motif # 39929).

### Chromatin immunoprecipitation
ChIP assays were performed essentially as described previously (Partridge et al, 2007), with the substitution of bead beating for enzymatic cell disruption methods. Bead beating was performed for 2 min at ambient temperature using a Biospec Products mini beadbeater. Buffers used for anti-Flag ChIPs did not contain sodium deoxycholate.
Ab for H3K9me2: Mouse monoclonal (Abcam ab1220). Ab for H3K9Ac: Rabbit polyclonal against histone H3 aa 4–14 K9ac (Millipore #07-352). Ab for H3K14Ac: Rabbit polyclonal against histone H3 aa 9–18 K14ac (Millipore#07-353). Ab for Flag: Mouse monoclonal M2 anti-Flag (Sigma #F3165). Ab for Swi6: Rabbit polyclonal Thermo Scientific PA 1-497.

### Chromosome segregation assays

Rates of chromosome missegregation were obtained as previously described (Petrie *et al*, 2005).

### Fluorogenic peptide deacetylation assay

Lysine deacetylase activity of Sir2 and other enzymes was evaluated using a fluorogenic peptide substrate (Fluor-de-Lys green HDAC drug discovery kit: BML-AK530-0001, Enzo Life Sciences). Please see Supplementary data for additional information.

### $^{32}$P NAD$^+$ hydrolysis assay for Sir2 deacetylase activity

$^{32}$P NAD$^+$ hydrolysis assays were performed essentially according to the method of (Tanny and Moazed, 2001). Please see Supplementary data for exact conditions.

### Supplementary data

Supplementary data are available at *The EMBO Journal* Online (http://www.embojournal.org).

# Acknowledgements

We thank Lorraine Pillus and Robin Allshire for strains; Keith Gull for anti-TAT1 antibody; Richard Festenstein for H3K4Ac Ab; Tony Carr, Kathy Gould, Susan Forsburg, and Rohinton Kamakaka for vectors; and the Hartwell Center of St Jude Children's Research Hospital for peptide synthesis and DNA sequencing. Thanks to Paul Brindle, Thomas Schalch, Jim Ihle, and Kevin Creamer for thoughtful comments. Funding was provided by NIH R01 GM084045 to JFP, Cancer Center support grant CCSG 2 P30 CA21765 (St Jude), and the American Lebanese Syrian Associated Charities (ALSAC) of St Jude Children's Research Hospital.

*Author contribution*: Project was conceived by JFP. BJA and JFP designed the research; BJA, GJ, RKY, SS, BRL, and JFP performed the research; JFP and BJA wrote the paper, and all authors reviewed the paper.

# Conflict of interest

The authors declare that they have no conflict of interest.

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
