## [Review Process File · The EMBO Journal]

Manuscript EMBO-2013-84738

Sir2 is required for Clr4 to initiate centromeric heterochromatin assembly in fission yeast

Benjamin J. Alper, Godwin Job, Rajesh K. Yadav, Sreenath Shanker, Brandon R. Lowe and Janet F. Partridge

Corresponding author: Janet F. Partridge, St. Jude Children's Research Hospital

Review timeline:

Submission date:	08 February 2013
Editorial Decision:	28 February 2013
Revision received:	15 May 2013
Editorial Decision:	23 May 2013
Accepted:	23 May 2013

Transaction Report:

Editor: Anke Sparmann

1st Editorial Decision

28 February 2013

Thank you again for submitting your manuscript (EMBOJ-2013-84738) to our editorial office. It has now been seen by two referees and their comments are provided below.

Both reviewers appreciate your study and are in general supportive of publication in The EMBO Journal. Nevertheless, they suggest several important experiments that need to be included in order to strengthen your conclusions, and emphasize that a significant revision of the manuscript will be required.

Given the comments provided, I would like to invite you to submit a suitably revised manuscript to The EMBO Journal that attends to the raised concerns in full. I should add that it is our policy to allow only a single major round of revision and that it is therefore important to address all criticism at this stage. Please do not hesitate to contact me should any particular point require further clarification.

When preparing your letter of response to the referees' comments, please bear in mind that this will form part of the Review Process File, and will therefore be available online to the community. For more details on our Transparent Editorial Process, please visit our website: <http://www.nature.com/emboj/about/process.html>

Thank you for the opportunity to consider your work for publication.
I look forward to your revision.

 REFEREE COMMENTS

Referee #1

Heterochromatin in the fission yeast has been serving as a model system to study the establishment, maintenance and inheritance of this well-conserved chromatin structure. It involves RNAi-dependent and -independent pathways, whose components also include histone methyltransferases (HMT) and histone deacetylases (HDAC), etc. The interplay between these factors contributes synergistically to the dynamics of heterochromatin. Such interplay varies depending on the different stages and locations of heterochromatin, yet its underlying mechanism remains largely unknown. Here, Alper et al. presented the correlation between two highly-conserved enzymes, Sir2 (HDAC) and Clr4 (HMT). They showed that Sir2 and its deacetylases activity are required for Clr4 recruitment, methyltransferases activity and hence heterochromatin establishment at certain stage and location. Therefore, this study should appear to a broad interest.

Such requirement of Sir2 and its deacetylases activity to promote Clr4 activity is evident at subtelomeric heterochromatin. However, at pericentromeric heterochromatin, the authors found that Sir2 is only imperative for de novo establishment but not for maintenance. And how Sir2 promotes Clr4 activity still needs clarification. To address this question, they started from testing the substrate specificity of Sir2 *in vitro*. Surprisingly, the preferred substrates, H3K4Ac and H4K16Ac, did not affect heterochromatin *in vivo* when mutated, while the weaker substrates (H3K9Ac and H3K14Ac) do. Because H3K9A mutation also abolishes the target for Clr4 recruitment and activity, they used H3K14A mutant to demonstrate that lack of deacetylation at H3K14Ac reduces Clr4 recruitment, thus providing one possible link between the two enzymes. Further, they tested such correlation at an euchromatic region by tethering Sir2 to the chromatin and inducing H3K9 methylation (by Clr4). This manuscript is well organized; however, there are several major points need to be addressed:

1. H3K14Ac is also the substrate of another HDAC, Clr3. Does Clr3 overlap or compete with Sir2's function? If it is redundancy, can it explain the intermediate phenotypes observed in the enzymatically inactive sir2 mutant (Sir2N247A) (Fig 4A, Fig 7C)? If lack of deacetylation at H3K14Ac reduces Clr4 recruitment, can it be phenocopied by H3K14R mutation which mimics the charge of de-acetylated lysine?
2. The discrepancy between the *in vitro* substrate specificity and corresponding mutant phenotype is confusing. It will help to test if such substrate specificity is true *in vivo*, e.g. test the change of amount of those substrates in sir2 mutant compared to WT.
3. How about the level of H3K4Ac and H4K16Ac at the Sir2-tethering location (Fig 7)? Could it be consistent with the *in vitro* substrate specificity since the redundancy of other pathways at pericentromeric heterochromatin does not exist in euchromatic region?
4. Please comment on why Sir2 is required for de novo establishment but not for maintenance.

Minor points:

1. The running title "Sir2 directs centromeric heterochromatin assembly" is a bit overstatement.

Referee #2

Fission yeast provides a good model system to analyze molecular mechanism of heterochromatin formation and function. In this manuscript, authors analyzed a role of NAD⁺-dependent deacetylase Sir2 in heterochromatin formation, which has not been established yet. Authors first found that Sir2 was necessary for de novo heterochromatin formation. Authors identified new histone targets for Sir2 and showed that the H3K14 acetylation (H3K14Ac) was important for recruitment of Clr4, H3K9 specific methyltransferase. Since artificial tethering of Sir2 to euchromatin promotes H3K14 deacetylation and recruits Clr4 activity, authors claimed Sir2-dependent H3K14 deacetylation functions upstream of Clr4 recruitment for heterochromatin assembly.

I feel the conclusion is interesting and has enough impact to attract the readers of EMBO Journal. However, in the same time, I feel that important experiments supporting their conclusion are still missing.

Major points

1. Authors measured the level of H3K9Ac in Figures 1, 2 and 3, but did not measure the level of H3K14Ac. Since main point of this paper is a role of Sir2-dependent H3K14-deacetylation in heterochromatin assembly, authors should analyze H3K14Ac in these experiments. Especially, analysis of H3K14Ac in Clr4-reintegration experiment (Figures 3 and 4) is important to confirm that requirement of H3K14 deacetylation in de novo heterochromatin formation.
2. In the same context as that described in the major point 1, Clr4-reintegration experiments should be done using H3K14A mutant. I believe heterochromatin formation can be monitored by measuring the level of Swi6 in this mutant.
3. Authors suggested that Sir2-requires Clr4 via H3K14-deacetylation in a RNAi-independent manner. Previously, authors' group demonstrated that existence of RNAi-independent H3K9-methylation using Clr4-reintegration experiments with *rna1-clr4* double mutants (Shanker et al. 2010, Plos Genet). Thus, the similar experiments using *rna1-clr4-sir2* triple mutants would show the requirement of Sir2 in the RNAi-independent H3K9-methylation.
4. It is surprising that H3K9me was not introduced in *sir2Δclr4Δ* to *clr4+* cells. It indicates that somehow RNAi system could not introduce H3K9me in the absence of Sir2 even though enough siRNA was produced. Which step was affected in the absence of Sir2? Since RITS recruits Clr4 via Stc1 in RNAi dependent system, it is worth to try to analyze localization of RITS and/or Stc1 in this condition.
5. Since the decrease of H3K14Ac was marginal in Figure 4 E, it is hard to say that H3K14 deacetylation is required for H3K4 methylation. Analysis of the effect of Sir2-tethering in H3K14A mutant would clearly indicate the requirement. The level of H3K9 methylation would be monitored by ChIP analysis of Clr4 and/or Swi6.

Minor points

1. In all *clr4* add back experiments, authors should confirm the expression level of *clr4* by RT-PCR or western blot to avoid the possibility of the deficient expression levels of "add backed *clr4+*" gene.
2. To emphasize the importance of Sir2 for de novo heterochromatin formation, authors should conduct *sir2+* (*sir2N247A*) add back experiment to "*sir2Δ clr4Δ* to *clr4+*" strain.
3. In Sir2 tethering experiments, authors should include Swi6 ChIP. In addition, authors should test RNAi dependency of the artificial heterochromatin.
4. It is possible that tethering of Sir2 represses transcription without heterochromatin formation, since deacetylation of histone generally correlate with gene silencing and the observed H3K9me level is relatively low. Inclusion of *clr4* mutant in the experiments would help to clarify this issue.

1st Revision - authors' response

15 May 2013

Detailed response to the reviewers:

We thank the reviewers for their helpful suggestions which we believe have strengthened our paper. Below we have provided a detailed response to the comments.

Referee #1

Heterochromatin in the fission yeast has been serving as a model system to study the establishment, maintenance and inheritance of this well-conserved chromatin structure. It involves RNAi-

dependent and -independent pathways, whose components also include histone methyltransferases (HMT) and histone deacetylases (HDAC), etc. The interplay between these factors contributes synergistically to the dynamics of heterochromatin. Such interplay varies depending on the different stages and locations of heterochromatin, yet its underlying mechanism remains largely unknown. Here, Alper et al. presented the correlation between two highly-conserved enzymes, Sir2 (HDAC) and Clr4 (HMT). They showed that Sir2 and its deacetylases activity are required for Clr4 recruitment, methyltransferases activity and hence heterochromatin establishment at certain stage and location. Therefore, this study should appear to a broad interest.

Such requirement of Sir2 and its deacetylases activity to promote Clr4 activity is evident at subtelomeric heterochromatin. However, at pericentromeric heterochromatin, the authors found that Sir2 is only imperative for de novo establishment but not for maintenance. And how Sir2 promotes Clr4 activity still needs clarification. To address this question, they started from testing the substrate specificity of Sir2 in vitro. Surprisingly, the preferred substrates, H3K4Ac and H4K16Ac, did not affect heterochromatin in vivo when mutated, while the weaker substrates (H3K9Ac and H3K14Ac) do. Because H3K9A mutation also abolishes the target for Clr4 recruitment and activity, they used H3K14A mutant to demonstrate that lack of deacetylation at H3K14Ac reduces Clr4 recruitment, thus providing one possible link between the two enzymes. Further, they tested such correlation at an euchromatic region by tethering Sir2 to the chromatin and inducing H3K9 methylation (by Clr4). This manuscript is well organized; however, there are several major points need to be addressed:

1. H3K14Ac is also the substrate of another HDAC, Clr3.

A. Does Clr3 overlap or compete with Sir2's function?

Response: We have tested this idea by combining deletion mutants in *clr3* and *sir2* and testing for silencing at centromeres. We found that Sir2 and Clr3 do appear to have overlapping function in centromeric heterochromatin maintenance, as the combined loss of both genes had a more severe impact on silencing than loss of either gene alone. This data has been included in Figure 2F.

B. If it is redundancy, can it explain the intermediate phenotypes observed in the enzymatically inactive *sir2* mutant (Sir2N247A) (Fig 4A, Fig 7C)?

Response: We have generated compound mutants of *sir2N247A* and *clr3* deletion and tested for silencing at centromeres. We found that, as for *clr3D sir2D*, there was loss of silencing in *clr3D sir2N247A* strains. This data has been included as Figure 2G. Given this result, we have gone on to monitor Swi6 recruitment (heterochromatin) at centromeres in the compound mutant strains. We found that Swi6 recruitment to centromeres was somewhat reduced in cells lacking *clr3* or *sir2*, but was completely lost from centromeres in *sir2Dclr3D* background and in the *sir2N247Aclr3D* background, with no enrichment of Swi6 above levels in a *clr4* deletion strain (this is shown in Figure 2H). Since heterochromatin maintenance is lost in the *clr3D sir2N247A* double mutant, we could not assess for redundancy during heterochromatin establishment by the *clr4* withdrawal and reintegration system. It is possible that the enzymatically inactive *sir2* provides a platform for Clr3 function during heterochromatin establishment, and we have included discussion of this possibility.

C. If lack of deacetylation at H3K14Ac reduces Clr4 recruitment, can it be phenocopied by H3K14R mutation which mimics the charge of de-acetylated lysine?

Response: We transformed H3K14R *clr4D* strains with episomal genomic Flag-Clr4 expressing plasmids, and observed by western analysis that the Flag-Clr4 is stably expressed in this genetic background (Figure S7C). We performed anti-Flag ChIP experiments and saw that like H3K14A, H3K14R blocks recruitment of Flag-Clr4 to centromeres. We have included this data in Figure S7D. This result was not unexpected since the phenotype of K14R (loss of silencing at centromeres) resembles that of K14A (Mellone et al., Curr Biol, 2003). Putting this data together with our demonstration that Sir2 is required for deacetylation of H3K14 following *clr4*⁺ reintegration (Figure 3G), it suggests that in addition to deacetylation of K14, that the K14 moiety itself may play an important role in recruitment of the Clr4 complex to chromatin.

2. The discrepancy between the in vitro substrate specificity and corresponding mutant phenotype is confusing. It will help to test if such substrate specificity is true in vivo, e.g. test the change of amount of those substrates in *sir2* mutant compared to WT.

Response: As stated in the text, it has previously been demonstrated that Sir2 is required for reduction of global levels of H3K4Ac (Xhelmalce and Kouzarides, 2010). We have used two different antibodies against H3K4Ac from Richard Festenstein and from Novus biological, but have not been able to detect a band in westerns of wt and *sir2D* extracts from fission yeast that is specific to histone H3. We have also monitored global H4K16Ac changes, and found a very slight increase in levels of total H4K16 acetylation in *sir2D* strains compared with WT strains when normalized to tubulin as a loading control. This data is included as Figure S4D.

3. How about the level of H3K4Ac and H4K16Ac at the Sir2-tethering location (Fig 7)? Could it be consistent with the in vitro substrate specificity since the redundancy of other pathways at pericentromeric heterochromatin does not exist in euchromatic region?

Response: This is a good point, and we tried extensively to test this. During the course of these and other experiments, it became apparent that the simple interpretation that targeting Sir2 to the euchromatic reporter resulted in silencing was not the full story. There appears to be a high incidence of loss of the *ade6* reporter specifically when Sir2 is targeted to the locus, which confounds interpretations of experiments based on the *ade6* reporter. We have therefore removed the targeting experiments from the paper. We have replaced these experiments with new data demonstrating that de novo accumulation of H3K9me2 is blocked following overexpression of *clr4*⁺ in cells triply deficient for *clr4*, *dcr1* and *sir2*, whereas H3K9me2 accumulates at centromeres on overexpression of *clr4*⁺ in *clr4Ddcr1D* cells (Shanker et al., PLOS Genetics 2010). This data supports a critical role of Sir2 in providing the correct environment for Clr4 function (see new Figure 7).

4. Please comment on why Sir2 is required for de novo establishment but not for maintenance.

Response: As outlined above (point 1), our new data shows that Sir2 and Clr3 have overlapping roles in heterochromatin maintenance. We present in this manuscript that Sir2 is required for heterochromatin establishment when using the sensitized genetic background of a *clr4* deletion followed by *clr4*⁺ reintegration. In this situation, the requirement for Sir2 function in recruiting Clr4 is revealed possibly because the normal redundancy with Clr3 is abrogated since the H3K9me2 mark is absent in *clr4D* strains, and Clr3 may depend on H3K9me2 for its recruitment to centromeres via targeting through the Chp2 chromodomain binding to H3K9me2.

Minor points:

1. The running title "Sir2 directs centromeric heterochromatin assembly" is a bit overstatement.

Response: We have modified the running title to "Sir2 regulates centromeric heterochromatin assembly"

Referee #2

Fission yeast provides a good model system to analyze molecular mechanism of heterochromatin formation and function. In this manuscript, authors analyzed a role of NAD⁺-dependent deacetylase Sir2 in heterochromatin formation, which has not been established yet. Authors first found that Sir2 was necessary for de novo heterochromatin formation. Authors identified new histone targets for Sir2 and showed that the H3K14 acetylation (H3K14Ac) was important for recruitment of Clr4, H3K9 specific methyltransferase. Since artificial tethering of Sir2 to euchromatin promotes H3K14 deacetylation and recruits Clr4 activity, authors claimed Sir2-dependent H3K14 deacetylation functions upstream of Clr4 recruitment for heterochromatin assembly.

I feel the conclusion is interesting and has enough impact to attract the readers of EMBO Journal. However, in the same time, I feel that important experiments supporting their conclusion are still missing.

Major points

1. Authors measured the level of H3K9Ac in Figures 1, 2 and 3, but did not measure the level of H3K14Ac. Since main point of this paper is a role of Sir2-dependent H3K14-deacetylation in heterochromatin assembly, authors should analyze H3K14Ac in these experiments. Especially, analysis of H3K14Ac in Clr4-reintegration experiment (Figures 3 and 4) is important to confirm that requirement of H3K14 deacetylation in de novo heterochromatin formation.

Response: We have performed H3K14Ac ChIP in the *clr4*⁺ reintegration experiment. Our data shows that levels of H3K14Ac are increased in *clr4D* strains, but are suppressed to WT levels on reintegration of *clr4*⁺. Importantly, reintegration of *clr4*⁺ into *sir2Dclr4D* cells does not lead to a reduction in H3K14Ac levels, consistent with a role of Sir2 in deacetylating H3K14. This data is included as Figure 3G.

2. In the same context as that described in the major point 1, Clr4-reintegration experiments should be done using H3K14A mutant. I believe heterochromatin formation can be monitored by measuring the level of Swi6 in this mutant.

Response: Strains expressing the H3K14A mutant show delocalization of GFP-Swi6 when assessed in living cells (Mellone et al., Curr Biol, 2003) and accumulation of centromeric transcripts (Fig. 6A and B). We performed *clr4*⁺ reintegration experiments in this background, and reintegrant strains show constitutively high levels of centromeric transcripts, as expected. This is shown in Figure S6. We have attempted Swi6 ChIP in strains expressing the H3K14A mutant, but as indicated from the localization data, Swi6 recruitment to centromeres is lost in the H3K14A mutant background, similar to *clr4D* strains (shown in Figure S7B). We therefore did not attempt Swi6 ChIPs following reintegration of *clr4*⁺ into *clr4D* H3K14A mutant background.

3. Authors suggested that Sir2-recurites Clr4 via H3K14-deacetylation in a RNAi-independent manner. Previously, authors' group demonstrated that existence of RNAi-independent H3K9-methylation using Clr4-reintegration experiments with *rna1-clr4* double mutants (Shanker et al. 2010, Plos Genet). Thus, the similar experiments using *rna1-clr4-sir2* triple mutants would show the requirement of Sir2 in the RNAi-independent H3K9-methylation.

Response: This is an interesting point and we thank the reviewer for this suggestion. We generated *clr4D sir2D dcr1D* strains and transformed with episomal genomic *clr4*⁺ expression vector, and monitored whether H3K9me could accumulate at centromeres as we had seen previously in *dcr1D clr4D* strains. We found that cells lacking *sir2 dcr1* and *clr4* were unable to support de novo deposition of H3K9me2 at centromeric dg or dh repeats following overexpression of *clr4*⁺ (shown in new Figure 7A). Thus the RNAi-independent H3K9 methylation is dependent on Sir2. We have extended and confirmed these findings by also monitoring Swi6 recruitment in these strains, and similarly see a complete failure to recruit Swi6 to centromeres in the triple mutant cells on overexpression of *clr4*⁺, whereas Swi6 is enriched at centromeres of *dcr1Dclr4D* cells on overexpression of *clr4*⁺. This new data is included in Figure 7B of the revised manuscript. We also monitored *clr4* transcript levels in these experiments, and show in Figure S9C, that the loss of H3K9me2 in the triple mutant cells cannot be attributed to a reduced level of *clr4* expression from the vector in these cells.

4. It is surprising that H3K9me was not introduced in *sir2Δclr4Δ* to *clr4*⁺ cells. It indicates that somehow RNAi system could not introduce H3K9me in the absence of Sir2 even though enough siRNA was produced. Which step was affected in the absence of Sir2? Since RITS recruits Clr4 via Stc1 in RNAi dependent system, it is worth to try to analyze localization of RITS and/or Stc1 in this condition.

Response: We have seen this type of situation previously. In establishment assays performed with Chp1 chromodomain mutants that are only slightly weakened for their in vitro binding activity for H3K9me2, we saw a failure in heterochromatin establishment on reintegration of *clr4*, in spite of the presence of normal levels of siRNAs (Schalch et al Mol Cell 2009). To address the reviewer's question, we performed Chp1 ChIPs, and found that whereas Chp1 was efficiently recruited to centromeres following reintegration of *clr4*⁺ into *clr4D* null cells, that only very low levels of Chp1 accumulated in *clr4*⁺ reintegrant *sir2Dclr4D* cells (shown in Figure 3H). This would suggest a defect in recruitment of RITS in cells exhibiting loss of H3K9me2, consistent with our earlier studies.

5. Since the decrease of H3K14Ac was marginal in Figure 4 E, it is hard to say that H3K14 deacetylation is required for H3K4 methylation. Analysis of the effect of Sir2-tethering in H3K14A mutant would clearly indicate the requirement. The level of H3K9 methylation would be monitored by ChIP analysis of Clr4 and/or Swi6.

Response: We performed Sir2-tethering experiments in the H3K14A background.

During analysis of these strains, and in other experiments using the targeted locus, it became apparent that the simple interpretation that targeting Sir2 to the euchromatic reporter resulted in silencing was not the full story. There appears to be a high incidence of loss of the *ade6* reporter when Sir2 is targeted to the locus in several genetic backgrounds, which confounds interpretations of experiments based on the *ade6* reporter. We have therefore removed the targeting experiments from the paper. In addition, unfortunately, as mentioned above (Ref 2 point 2), Swi6 cannot ChIP in the H3K14A mutant background (Figure S7B). We have also shown that Clr4 cannot be recruited to chromatin in H3K14A strains (Figure 6H), therefore we could not assess heterochromatin formation in the H3K14A mutant. However, we would suggest that H3K9 is unlikely to be methylated because of lack of localization of Clr4 in the H3K14A background.

Minor points

1. In all *clr4* add back experiments, authors should confirm the expression level of *clr4* by RT-PCR or western blot to avoid the possibility of the deficient expression levels of "add backed *clr4*+" gene.

Response: We have used RT-PCR to check levels of *clr4*⁺ expression following its reintegration into *clr4D* or *clr4D**sir2D* strains, and see no evidence of a reduction in *clr4*⁺ transcript levels (Figure S2A). In addition, we tested expression of Flag-Clr4 proteins in histone mutant backgrounds (Figure 6G, Figure S7C), and again see no evidence for a difference in levels of expression. Additionally, in the experiment where *clr4*⁺ is expressed in the triple mutant background (Figure 7), we see no evidence for downregulation of *clr4*⁺ expression (shown in Figure S9C).

2. To emphasize the importance of Sir2 for de novo heterochromatin formation, authors should conduct *sir2*⁺ (*sir2N247A*) add back experiment to "*sir2Δ clr4Δ* to *clr4*+" strain.

Response: We have performed such an "add back" experiment- where we episomally expressed *sir2*⁺ in the *sir2D**clr4D* to *clr4*⁺ strain background. We saw full complementation of centromeric silencing (Figure S2B and C).

3. In Sir2 tethering experiments, authors should include Swi6 ChIP. In addition, authors should test RNAi dependency of the artificial heterochromatin.

Response: We performed Swi6 ChIP in the Sir2-tethering experiments, and also tested RNAi and Clr4 dependency (point 4 below) of the silencing. During the course of these experiments, it became clear that the simple interpretation that targeting Sir2 to the euchromatic reporter resulted in silencing was not the full story. There appears to be a high incidence of loss of the *ade6* reporter specifically when Sir2 is targeted to the locus, which confounds interpretations of experiments based on the *ade6* reporter. We have therefore removed the targeting experiments from the paper.

4. It is possible that tethering of Sir2 represses transcription without heterochromatin formation, since deacetylation of histone generally correlate with gene silencing and the observed H3K9me level is relatively low. Inclusion of *clr4* mutant in the experiments would help to clarify this issue.

Response: See response to point 3 above

2nd Editorial Decision

23 May 2013

Thank you for submitting your revised manuscript for our consideration. I am very pleased to inform you that in light of the re-review comments from one of the original referees (provided below), we are happy to accept your paper for publication in The EMBO Journal.

Please note that in order to be able to start the production process, we would require you to complete and sign the linked license agreements (see below).

Thank you very much for your contribution to The EMBO Journal!

I look forward to seeing your article published.

REFEREE COMMENTS

Referee #2

The revised manuscript adequately answered all the issues raised by me (referee #2). My main criticism to the previous version was that the results showing the importance of H3K14 deacetylation in heterochromatin formation was not enough. Now the revised manuscript satisfactorily indicates the importance of H3K14 deacetylation with new experiments.